# Clickbait Detection Using Deep Recurrent Neural Network

Abdul Razaque [1,*], Bandar Alotaibi [2,3,*], Munif Alotaibi [4,*], Shujaat Hussain [5], Aziz Alotaibi [6] and Vladimir Jotsov [1,7]

1 Department of Computer Engineering and Cybersecurity, International Information Technology University, Almaty 050000, Kazakhstan
2 Department of Information Technology, University of Tabuk, Tabuk 47731, Saudi Arabia
3 Sensor Networks and Cellular Systems (SNCS) Research Center, University of Tabuk, Tabuk 47731, Saudi Arabia
4 Department of Computer Science, Shaqra University, Shaqra 11961, Saudi Arabia
5 Department of Computing, National University of Computer and Emerging Sciences, G-9/4, Islamabad 44000, Pakistan; shujaat.hussain@nu.edu.pk
6 Department of Computer Science, College of Computers and Information Technology, Taif University, P.O. Box 11099, Taif 21944, Saudi Arabia; azotaibi@tu.edu.sa
7 Department of Information Technology, University of Library Studies and Information Technologies, 1784 Sofia, Bulgaria; v.jotsov@unibit.bg
* Correspondence: a.razaque@iitu.edu.kz (A.R.); b-alotaibi@ut.edu.sa (B.A.); munif@su.edu.sa (M.A.)

**Abstract:** People who use social networks often fall prey to clickbait, which is commonly exploited by scammers. The scammer attempts to create a striking headline that attracts the majority of users to click an attached link. Users who follow the link can be redirected to a fraudulent resource, where their personal data are easily extracted. To solve this problem, a novel browser extension named ClickBaitSecurity is proposed, which helps to evaluate the security of a link. The novel extension is based on the legitimate and illegitimate list search (LILS) algorithm and the domain rating check (DRC) algorithm. Both of these algorithms incorporate binary search features to detect malicious content more quickly and more efficiently. Furthermore, ClickBaitSecurity leverages the features of a deep recurrent neural network (RNN). The proposed ClickBaitSecurity solution has greater accuracy in detecting malicious and safe links compared to existing solutions.

**Keywords:** clickbait; security; malicious links; non-malicious links; deep learning; RNN



## 1. Introduction

In the last few decades, individuals have been increasingly involved in various activities on social networks [1]. Social networks are utilized by roughly half the global population [2]. Prior to the emergence of the Internet, headlines were displayed in banners and newspapers [3]. With the evolution of various Internet services, such as e-commerce [4,5], clickbait has become prevalent. The more that website visitors engage with clickbait, the greater the satisfaction of the company with the demand for the advertised product. On the other hand, if clickbait does not attract many users, this proves that individuals are not interested in this product, and the company might not make a profit [6].

Regrettably, attackers have started to extensively exploit this valuable utility to fool website users into visiting harmful Internet resources, where their private information can be hijacked [7]. Clickbait is considered one of the most harmful types of malicious activities since it can contain not only malicious links [8] but also executable code [9]. Any user who visits a certain website is a potential phishing target [10]. Moreover, prohibited Internet resources can be visited through a malicious link that redirects website visitors to these resources [11]. Some sites remotely utilize the user's computer power for mining [12]. A harmful link is designed so that naive users cannot distinguish it from a link that contains a regular advertisement [13]. Thus, clickbait is a serious threat that must be addressed. A standard framework of clickbait detection is shown in Figure 1.

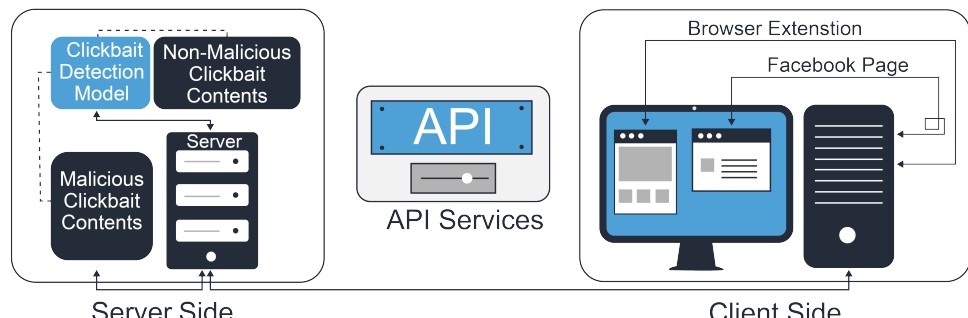

**Figure 1.** Simple model of clickbait detection.

To resolve this issue, we designed a novel browser extension that helps the user to recognize malicious links in clickbait. Analogs of such extensions already exist (e.g., AdBlock, AdFender, and AdGuard) [14]. However, existing approaches block the majority of ads from distinct sources [15], and hence, both harmful and useful clickbait links are blocked. In contrast, our proposed novel browser extension detects harmful clickbait, provides users with all possible information, and offers recommendations for further improvement and action.

## 1.1. Research Contributions

Motivated by the room for improvement and shortcomings in existing approaches, this paper's contributions can be summarized as follows:

- A novel browser extension is proposed that analyzes links attached to clickbait while using less memory.
- The proposed ClickBaitSecurity is capable of efficiently distinguishing between legitimate and illegitimate links by accurately using an RNN.
- The novel proposed ClickBaitSecurity integrates the features of binary and domain rating check algorithms for a faster and more efficient search process.

## 1.2. Paper Organization

This article is organized as follows. Section 2 identifies the problem and discusses its importance, and related work is described in Section 3. The system model for clickbait is presented in Section 4. The scanning process of the proposed extension is detailed in Section 5, and the deep recurrent neural network for malicious content detection is presented in Section 6. The experiment and its results are presented in Section 7. The advantages and limitations of the proposed method are discussed in Section 8, and the conclusion of the paper is introduced in Section 9.

## 2. Problem Identification and Importance

Users can be exposed to major privacy threats through mechanisms such as clickbait. Harmful clickbait can lure naive users into believing that they are about to click a benign advertisement related to their favorite products when they are actually about to encounter a major threat. Since the Internet emerged, cybercriminals have continuously found ways to utilize network resources to target users. Organizations can face considerable losses if an intruder penetrates their websites to execute illegitimate actions. Unfortunately, this issue has evolved over time as hackers have developed new tools with aggressive capabilities. Internet browsers have various vulnerabilities that make them targets of cybercriminals. Since 2016, about 50% of cyberattacks have targeted Internet browsers. Moreover, at least 77% of the attacks have been launched by utilizing malicious links [16]. These threats arise because many users lack sufficient awareness to comprehend and analyze Internet resources. Furthermore, new malicious activities emerge on a daily basis. Therefore, the following recommendations are introduced:

- Educate naive users about threats that they might encounter on the Internet by creating more services.
- Publish a list of well-known fraudulent Internet resources in order to warn users.
- Protect users from suspicious links that appear in advertisements using intrusion detection/prevention tools.

The most useful recommendation with the greatest benefit is to protect users from suspicious links by deploying intrusion detection/prevention tools. Nevertheless, even experts can make mistakes and click on malicious links.

## 3. Related Work

This section introduces key aspects of existing work. A suitable approach to decreasing the number of targeted advertisements is to filter user posts on social media. The filtering utility is used with the help of special firewalls to prevent browsers from reading private information [17]. This method is distinctive because it prevents personal data from being revealed and decreases the number of targeted advertisements. However, filed advertisements might not be decreased when deploying this method.

Malicious links are blocked by utilizing whitelists and blacklists. White- and blacklists are built in order to prevent harmful Internet resources from affecting users [18]. Although employing white- and blacklists might facilitate the attainment of precise information about Internet resources, these lists might not include all available sources.

Establishing a set of rules can protect users from being targeted by well-known Internet tricks. Sahoo et al. [19] proposed valuable, simple rules that protect users from being targeted by malicious resources. For instance, the content of suspicious titles is checked, or scripts that belong to unverified sources are disabled. This approach provides some useful rules to avoid being victimized on the Internet; however, most Internet fraud can occur even if users follow the suggested rules.

Khan and Kim [20] utilized artificial intelligence to protect users from cyberattacks. This approach can effectively prevent Internet attacks by using machine learning. The proposed solution covers most cyberattacks when the machine learning algorithm is trained so that it can predict most attacks. Although machine learning can be an adequate solution to detect many attacks, it is challenging to cover all Internet attacks because new ones can emerge every day.

Currently, the lack of a universal solution that can be used to detect/prevent phishing is a serious issue. Thus, personal information may be protected against theft if Internet users follow certain rules. Arachchilage and Hameed [21] proposed specific rules that might be followed by users to avoid phishing. Some of these rules are: entering the website URL that users intend to visit and filtering email messages manually. Although authors provide some valuable rules to protect personal data from theft, they do not cover all phishing attempts.

To protect user data, some researchers presented a browser-based extension in order to block a certain amount of personal information. Mathur et al. [22] proposed a browser extension designed for Google Chrome. This extension is capable of protecting personal data from theft. The proposed method searches for and blocks active scripts on websites that can communicate to reveal user data. This method blocks all website advertisements. Therefore, this solution has a deficiency because benign/useful advertisements are also blocked.

Kaur et al. [23] proposed a deep learning-based approach to improve the performance of clickbait detection in terms of accuracy. This method combines both a convolutional neural network and long short-term memory (C-LSTM) to recognize more instances of clickbait on social media than existing methods. Moreover, this technique can detect headlines that contain malicious clickbait on various social media platforms. However, this method utilizes two deep learning algorithms that have high complexity, so the detection time and the computation cost might be high.

Zheng et al. [11] proposed a solution called lure and similarity for adaptive clickbait (LSAC) to detect clickbait. This method utilizes a combination of similarity and lure features.

The adaptive prediction utility is an important feature introduced by the authors. The authors created a Chinese clickbait to validate the proposed solution. This dataset consists of approximately 5000 media news items. This approach is based on a famous deep learning architecture known as the convolutional neural network. The proposed solution achieved high performance and was proven to be effective. However, the proposed method only considers media and news content, and the detection time of harmful/benign instances of clickbait is long.

Siregar et al. [24] proposed an approach based on recurrent neural network-bidirectional long short-memory to specify words that affect the scores of clickbait ads. Additionally, similarity information between the source and target is specified by utilizing a Siamese net. Subsequently, the convolutional neural network is used to recognize image embedding from a large amount of data, which adds complexity overhead to the introduced solution. The authors conducted experiments on 19,538 social media posts to evaluate their method. Although the proposed method achieved satisfactory accuracy, this solution is not designed for malicious clickbait detection and is limited to attribute similarity scoring and headline classification. Probierz et al. [25] proposed a method that balances data evaluation and classification of fake news forecasting. Natural language processing is employed to describe the text and title of the news. To address this issue, classical ensemble methods (CEMs) have been introduced.

All of the previously proposed approaches concentrate on performance improvement in terms of accuracy. However, they overlook the importance of detection time and memory usage. Thus, our approach considers these neglected factors (i.e., fast detection time and low memory usage) as well as accuracy. Table 1 compares and contrasts the existing methods. Additionally, Table 2 provides a detailed comparison of differences (low, medium, high, yes, no). In the comparison, the low was <95, medium was >95 && <98, and high was >98.

**Table 1.** Summary of the contributions of existing approaches.

| Method | Clickbait Detection Protocol | Features/Characteristics | Deficiencies |
|---|---|---|---|
| [17] | A protocol that uses a firewall to block malicious advertising | The ability to restrict malicious advertisements and prevent clients from revealing sensitive information | Incapable of discovering the filed advertisement |
| [18] | Blacklist and whitelist detection technique | Blocking suspicious Internet resources and the given information about Internet resources is delicate | Lack of completeness (i.e., it is not capable of providing a complete list of Internet resources). Additionally, it is not superior in terms of accuracy |
| [19] | Constructing a set of rules to protect clients from malicious resources | The capacity to prevent unverified sources by checking suspicious title content and disabling execution of scripts | Following the suggested rules does not prevent most Internet fraud |
| [20] | An approach based on artificial intelligence | The ability to detect various cybersecurity attacks using machine learning and predefined signatures | Ineffective in detecting new attacks |
| [21] | A rule-based method to detect/prevent clickbait and phishing | Some of the provided filtering rules are able to prevent phishing and clickbait | Not all phishing activities are covered by the suggested rules |
| [22] | A browser-based extension method to detect clickbait | The ability to find and block active scripts that can communicate with clients to reveal personal information | Useful advertisements are also blocked |
| [23] | Clickbait detection based on deep learning | High performance in terms of accuracy to detect more clickbait on social media. Moreover, it is able to detect headlines that consist of suspicious clickbait on several social media platforms | It is computationally complex because it combines two deep learning architectures (i.e., LSTM and CNN) |

**Table 1.** *Cont.*

| Method | Clickbait Detection Protocol | Features/Characteristics | Deficiencies |
|---|---|---|---|
| [11] | A convolutional neural network-based approach | It combines both similarity and lure features and achieved good accuracy using an adequately sized dataset | The detection time is long, and this method is limited to media and news content |
| [24] | An approach that consists of multiple stages | The accuracy of this approach is high compared to existing techniques | Limited to headline classification and attribute similarity scoring (i.e., it is not designed to detect malicious clickbait) |
| ClickBaitSecurity | Clickbait detection using deep recurrent neural network | Clickbait and source rating analysis and multilayered clickbait identification and search process. This approach can efficiently and accurately detect malicious content. It outperforms existing approaches in terms of accuracy, CPU consumption, memory usage, and link detection | It has some marginal deficiencies inherited from blockchain technology |

**Table 2.** Detailed comparison on differences.

| Method | Source Rating | Accuracy | Unverified Source Detection | False Positive Rate | False Negative Rate | Memory/CPU Consumption |
|---|---|---|---|---|---|---|
| [17] | No | Low | No | High | High | High |
| [18] | No | Low | Yes | Medium | High | Medium |
| [19] | No | Low | Yes | High | High | High |
| [20] | No | Medium | No | Medium | Medium | Medium |
| [21] | Yes | Low | No | Medium | Medium | Medium |
| [22] | No | Low | Yes | High | High | High |
| [23] | No | Low | Yes | High | High | High |
| [11] | No | Medium | No | Medium | Medium | High |
| [24] | No | Medium | No | Medium | Medium | High |
| ClickBaitSecurity | Yes | High | Yes | Low | Low | Low |

## 4. System Model

The system model of the new proposed ClickBaitSecurity extension consists of three elements: The first element is a user who has a personal computer with Internet access. The user's personal computer must comply with modern standards of use. The second element is a browser that must be updated to the latest version to comply with all of the latest security regulations. Additionally, the proposed extension for the analysis of clickbait must be installed in this browser. A small software module known as a browser extension can be utilized to customize the web browser. Various extensions are permitted by browsers, including extensions to manage cookies, block ads, and modify the user interface. The most popular browser, Google Chrome, has thousands of extensions that are available for free. The extension itself must be downloaded from the official Google store. The third element is a window by default for displaying messages from the browser, where all information about the analysis of the web link will be displayed. All of these elements are supported by the features of the recurrent neural network. Figure 2 shows the flow of an analysis process that touches on the three main elements described.

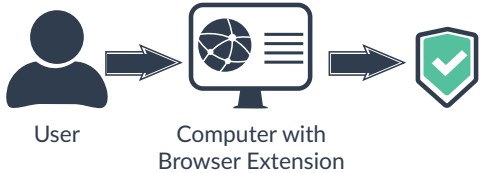

**Figure 2.** Flow of the analysis process.

When the user wants to use the system, the links appear on their system as depicted in Figure 3. If the user clicks the links and it might be malicious links to be the cause of the damage to private and critical information. Thus, the user has implemented algorithms and models to detect any possibility of malicious clickbait. The user initiates the activation process of the LILS algorithm in the message (2). When the LILS algorithm is activated, it tries to detect whether the source of the links is white-listed or black-listed shown in the message (3). The LILS has the support of the search and rating time calculation (SRTC) model that calculates the time spent for searching the black-listed and white-listed links. If the black-listed links are detected, then the user gets the results of detected links shown in the message (4). The black-listed links carry the malicious, suspended, and illegitimate web resources to launch potential threats. Further, the domains of the black-listed links are determined by activating the DRC algorithm shown in the message (5) that rates each domain based on the source of the links shown in the message (6). The domain rating is used to decide whether clickbait should be permitted or refused.

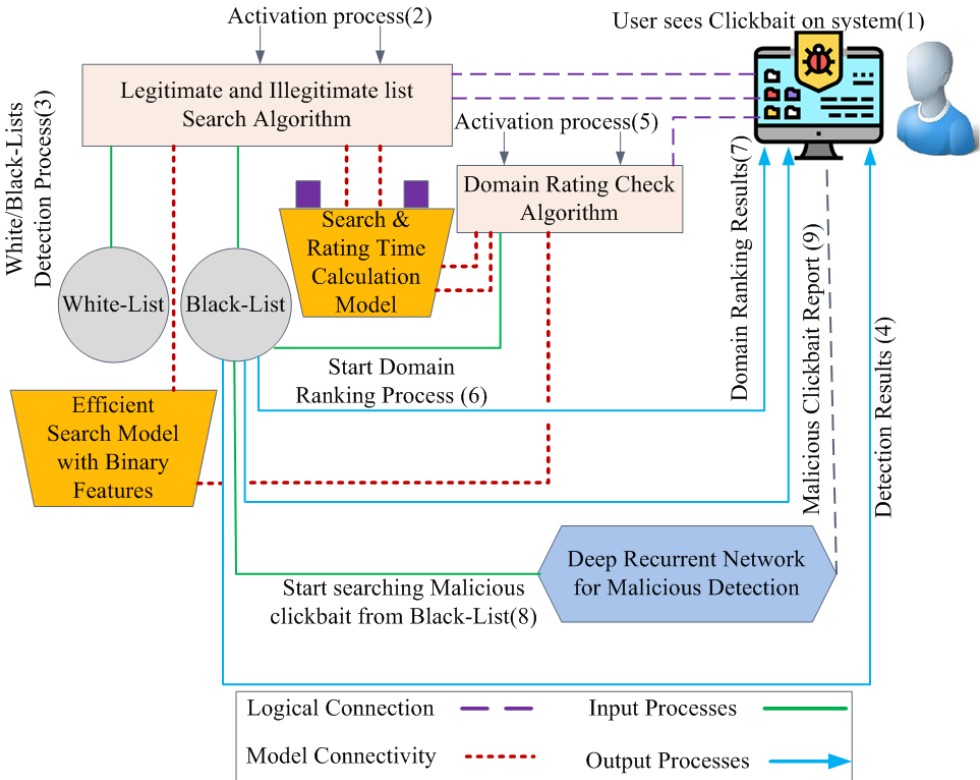

**Figure 3.** Process for detecting malicious clickbait.

The domains that have the highest ranking are informed to the user shown in the message (7). The DRC algorithm has also the support of the SRTC model to calculate time spent for rating the domains. The LILS and DRC algorithms leverage the features from the efficient search model for making an efficient searching process of detecting the black-listed and white-listed links and ranking the domains of the black-listed efficiently. The lower-ranked domains are checked to detect the possibility of malicious clickbait using

the recurrent neural network shown in the message (8). If the malicious clickbait is detected from the lower-ranked domains, the user is informed about this malicious activity. Thus, the user stops any potential threat by blocking the link.

The step-by-step procedure that supports these filters is depicted in Figure 4.

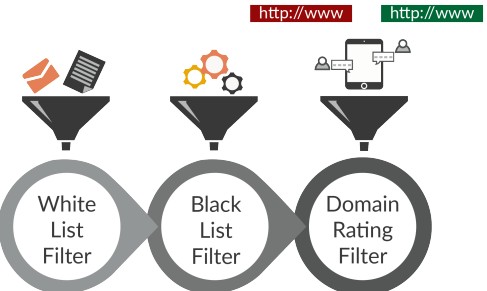

**Figure 4.** Link-filtering process for detecting malicious clickbait.

## 5. Proposed Extension Scanning Process

The proposed method is capable of detecting the described elements in an efficient manner. This solution is designed to ensure that the links that the user follows are safe to use. Our proposed extension focuses specifically on security checks. The process consists of three phases:

- Analysis and examination of legal websites;
- Examination of search process;
- Recurrent neural network for malicious content detection.

### 5.1. Analysis and Examination of Legal Websites

The process of analyzing clickbait is the most important phase since this is the main function of the presented browser extension. After this process, the user is notified as to whether the web resource is safe. To better explain the process of analyzing and checking a clickbait web resource, the process of analyzing clickbait is depicted in Figure 5. This figure shows the complete analysis process of checking the web link to establish its safety.

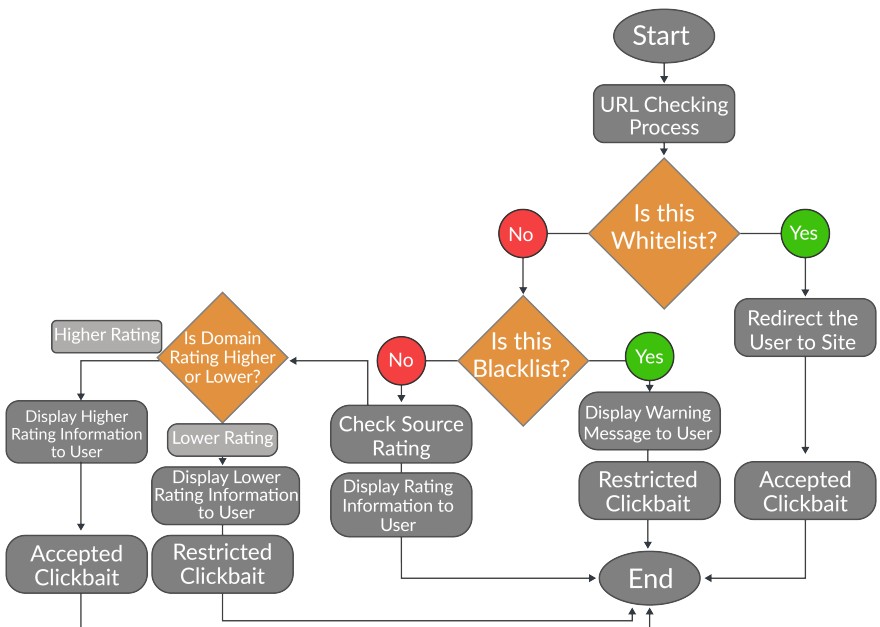

**Figure 5.** Process of analyzing clickbait.

To determine the white/blacklisted content of the clickbait, a legitimate and illegitimate list search algorithm is used. If we do not have accurate information about the web

resource after Algorithm 1, then the program proceeds to Algorithm 2. The domain rating check algorithm, which provides ranking of the domains.

---

**Algorithm 1** Legitimate and illegitimate list search algorithm.

---

**Input:** $S_U, B_L, W_L$ in
**Output:** $W_M$ *or* $O_M$ out
  1: **Initialization:** $\{S_U$: *Site URL;* $W_M$: *Warning message;* $O_M$: *Okay message;* $B_L$: *Blacklist;*
     $W_L$: *Whitelist;*$\}$
  2: **if** $S_U \in B_L$ **then**
  3:    **Show** $W_M$
  4: **end if**
  5: **if** $S_U \in W_L$ **then**
  6:    **Show** $O_M$
  7: **end if**
  8: **if** $C_K\ ! \in W_L$ *and* $S_U\ ! \in B_L$ **then**
  9:    **Show** $O_M$
10: **end if**

---

Algorithm 1 describes the process of determining link safety. In step 1, the variables are initialized for the safety determination process. The input and output processes are shown at the begining of the algorithm, respectively. Steps 2–4 check the blacklist to determine whether it contains the website URL. If the blacklist contains the URL, then the user is shown a warning message. Steps 5–7 check the whitelist to determine whether it contains the website URL. If the whitelist contains the URL, then the user receives a message indicating that the link is safe. In steps 8–10, the user is notified as to whether this website is new.

---

**Algorithm 2** Domain rating check algorithm

---

**Input:** $S_U, D_L, D_R$ in
**Output:** $W_M$ *or* $O_M$ out
    **Initialization:** $\{S_U$: *Site URL;* $W_M$: *Warning message;* $O_M$: *Okay message;* $D_R$: *Domain*
    *rating;*$\}$
  2: **if** $D_R < 1$ **then**
     **Show** $W_M$
  4: **end if**
     **if** $D_R > 1$ **then**
  6:    **Show** $O_M$
     **end if**

---

Algorithm 2 describes the process of determining link safety. In step 1, the variables are initialized for the safety determination process. The input and output processes are shown at the beginning of the algorithm, respectively. Steps 2–4 check the URL's domain rating. If the rating is less than or equal to 1, then a warning message is shown. In steps 5–7, if the domain rating is higher than 1, then the user receives a message that all is normal.

In this extension, the time allotted to return a message to the user about the web resource can be calculated. The best time for returning the message after the whitelist check step is calculated by:

$$T = T_s + T_W \tag{1}$$

where $T$ is the total time used by the program, $T_W$ is the time to check the whitelist, and $T_s$ is the time to send the request for accessing the web resource.

The best average time for returning a message during the blacklist check process can be calculated as:

$$T = T_s + T_B + T_W \tag{2}$$

where $T_B$ is the time to check the blacklist.

The worst average time for returning a message during the domain rating check can be calculated as:

$$T = T_s + T_B + T_W + T_D \tag{3}$$

where $T_D$ is the time to check the domain rating.

When additional analysis is required, the worst time is calculated by:

$$T = T_s + T_B + T_W + T_D + T_A \tag{4}$$

where $T_A$ is the time for advanced analysis.

### 5.2. Examination of Search Process

In our extension, we use binary search features. Because we use sorted lists of URLs, the binary search utilizes the sorted list of URLs to complete its mission. In the worst case, the binary search employs a comparison loop over many iterations, as shown in the following equation:

$$A_m = log_2^n + 1 \tag{5}$$

where $A_m$ is the worst case for the number of iterations, and $n$ is the number of elements in the array.

On average, assuming that each element is equally searched, a binary search calculates the number of iterations as follows:

$$A_m = (log_2 n + 1 - (x^{log_2 n+1} + 1) - 2 \times log_2 \frac{n}{n}) \tag{6}$$

Let us also analyze the best and worst cases of the binary search.

In the structure of the binary tree, an effective search is exemplified by a trajectory from the root of the tree to the target node. The initial iteration is given by:

$$I_N = (l + 1) \tag{7}$$

where $l$ is the path length, and $I_N$ is the initial iteration.

For an effective search, the average number of iterations can be calculated by:

$$T_N = 1 + \frac{I_N}{n} \tag{8}$$

where $T_N$ is the average number of iterations in an effective search.

As binary search features are optimal for searching with comparisons, this process is limited to calculating the minimum internal path length of the entire binary search with $n$ nodes, which can be demonstrated by:

$$M_l = \sum_{k=1}^{n} log_2 k \tag{9}$$

where $m_l$ is the minimum internal path length.

**Theorem 1.** *In a 7-element array, the program will have a minimum internal path that equals 254 when comparing all elements.*

**Proof.** Equation (9) is used to calculate the minimum internal path to obtain the solution. □

$$\sum_{k=1}^{7} log_2 k = 2 + 4 + 8 + 16 + 32 + 64 + 128 = 254$$

Ineffective searches can be performed by boosting the tree with external nodes. The average number of iterations in an ineffective search is represented by:

$$T'_N = \frac{E_N}{n+1}$$ (10)

where $T'_N$ is the average number of iterations in an ineffective search, and $E_N$ is the path length.

The equation can be substituted for $E_N$ in the equation for $T'_N$, and the worst case for unsuccessful searches is specified by:

$$T'_N = (log_2 n + 2 - 2^{log_2 n+1}) - \frac{(n+1)}{n}$$ (11)

This algorithm has an average search speed, which makes it an acceptable solution for a browser extension.

After the domain rating check and search algorithms have been executed, additional analysis of the web resource begins. The additional analysis consists of calculating various indicators. To determine whether a link is harmful in the additional analysis, we check the depth of the legal address. We obtain the depth indicator, which is calculated by:

$$h = \frac{1}{S_D}$$ (12)

where $h$ is the depth indicator, and $S_D$ is the number of subdomains.

When checking for security, the link is divided into main and domain parts. The importance of the main part is 75% (0.75), and the importance of the domain part is 25% (0.25). After that, the percentage of matches is calculated by:

$$C = 0.75 \times C_M + 0.25 \times \frac{C_D}{100}$$ (13)

where $C_M$ is the coincidence of the main part, $C_D$ is the coincidence of the domain part, and $C$ is the percentage of matches.

Based on the coincidence of the domain rating, where the importance of two parts equals 50% (0.5), the initial security check score is calculated as:

$$B_I = 0.5 \times S_C + 0.5 \times C_D$$ (14)

where $B_I$ is the base security indicator, and $S_C$ is the availability of a secure connection (if the site has a secure connection, then the parameter is set to 1; otherwise, it is 0).

Another parameter that is considered in the analysis is site visibility (Appendix A). This parameter can be calculated as:

$$V = D \times \sqrt{B}$$ (15)

where $V$ is the visibility parameter, $B$ is the number of backlinks that lead to the site and $D$ is the number of domains where these backlinks are located.

$$V = 2 \times \sqrt{14} = 7.48$$ (16)

Additional analysis also includes checking the site traffic parameter, which is calculated as:

$$A = \frac{I}{R_{lt}}$$ (17)

where $A$ is the attendance parameter, $l$ is the number of incoming users, and $R_{lt}$ is the resource lifetime.

Based on the site attendance, the site rating is calculated as:

$$S_R = \frac{A}{P} \tag{18}$$

where $S_R$ is the site rating, and $P$ is the number of pages in the Google search.

To store information, a dictionary collection is used, which performs well in adding, taking, and deleting operations that are calculated by:

$$O_B = log_1 D_n \tag{19}$$

where $O_B$ is the best time complexity, and $D_n$ is the number of elements in the dictionary.

$$O_A = log_2 D_n \tag{20}$$

where $O_A$ is average time complexity.

$$O_W = log_2 D_n + 1 \tag{21}$$

where $O_W$ is the worst time complexity.

Additionally, the browser extension checks certain data on the site that are attached to the clickbait. The extension determines the amount of executable code on the site, given by:

$$E_C = \sum_{k=1}^{n} J_k \tag{22}$$

where $E_I$ is the total amount of executed code, and $J$ is the detected JavaScript.

In the next step, we calculate the maliciousness index of the executable JavaScript code, which can be calculated by:

$$E_I = \sum_{k=1}^{n} J_k \times R \tag{23}$$

where $E_I$ is the index of executed code, and $R$ is a malicious link redirect parameter (if the code redirects to a malicious resource, then $R = 1$; if the code has suspicious logic, then $R = 0.5$, and $R = 0$ otherwise).

The extension uses an external sandbox to execute the JavaScript code and determine what the external program will produce. The time taken for this check is calculated by:

$$T_S = A_E \times T_E \tag{24}$$

where $T_S$ is the time for sandbox checking, $A_E$ is the number of rows in a single code, and $T_E$ is the time to check a single row.

The sandbox returns a sandbox indicator, which is calculated by:

$$S_I = \frac{W_M + M_M}{C_A} \tag{25}$$

where $S_I$ is the sandbox indicator, $W_M$ is the number of warning messages, $M_M$ is the number of malicious code messages, and $C_A$ is the total amount of checked code.

Finally, we calculate the complete indicator of malicious code, where the importance of two parts equals 50% (0.5), which is demonstrated by:

$$C_I = E_I \times 0.5 + W_I \times 0.5 \tag{26}$$

where $E_I$ is the index of executed code, and $C_I$ is the complete indicator of executed code.

## 6. Deep Recurrent Neural Network for Malicious Content Detection

Malicious content needs to be extracted from links. The malicious content from links is detected using an autoencoder. Autoencoders can check for hidden malicious content as well as new malicious content from the input. Let us assume that autoencoder $\beta$ is used to extract $\omega$. The malicious content from link $l$ is written as: $\{l = l^1, l^2, \dots, l^m\}$, where $l^m$ is the total amount of malicious content for each link.

Malicious content $C_m$ from link $l$ is deduced by the autoencoder. The formulation is as follows: $\beta e : l \rightarrow \omega$. The concealed malicious content is removed, and the link is rebuilt by decoder $\beta d$. The formulation is as follows: $\beta d : \omega \rightarrow l$, where $\omega$ is the content illustration for the link when using the autoencoder and auto-decoder.

The RNN is used as autoencoder $\beta e$, which learns the probability of detecting malicious links by being trained.

There is a hidden state $S_h$ and possible output to be activated on the total malicious content of the link; therefore, the input and output hidden states of the RNN are updated by each time step $t$. Time step $t$ is calculated by Equations (26) and (27) and depicted in Figure 6.

$$S_h(t) = f(\beta e)(W_i S_h(t) - 1, l^t) \tag{27}$$

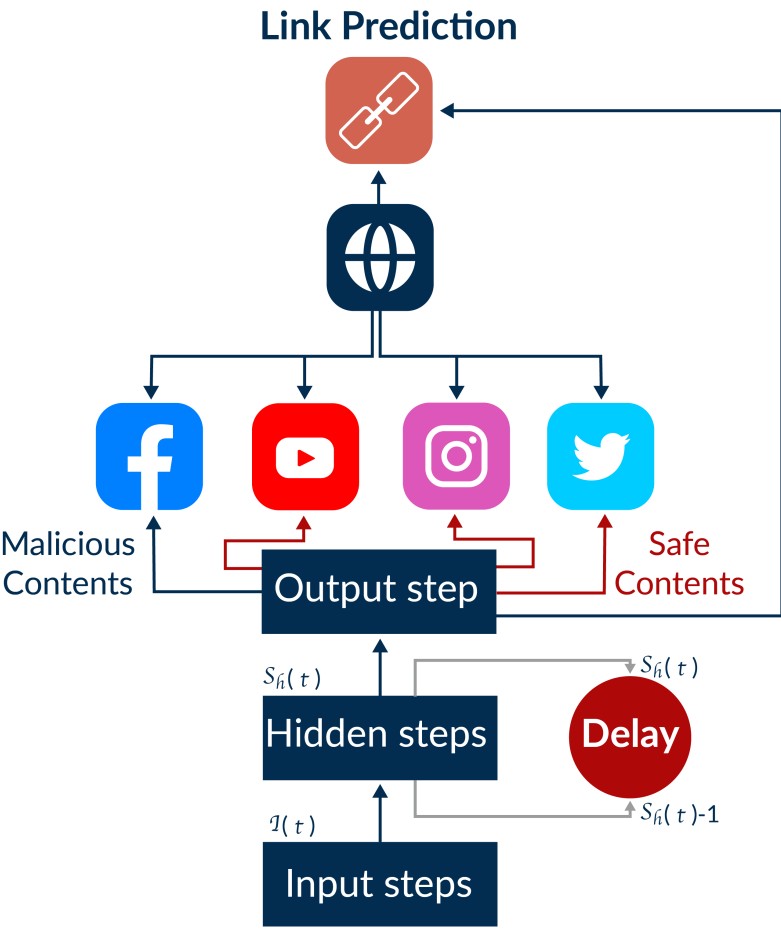

**Figure 6.** Recurrent neural network for malicious/safe content detection from links.

The final hidden state of the RNN is applied as content illustration $\omega$ for the link when it is accessed. The gated recurrent unit (GRU) is used to create the RNN. The GRU is identical to long short-term memory and supported by the forget gate. GRU is used because it requires fewer parameters and trains faster than LSTM in most scenarios.

Figure 7 depicts that an automatic encoder $\beta e$ receives the input link $I(t)$ from the GRU to determine the probability of malicious content. The sigmoid function is used to process the input to the next hidden state $S_h(t) - 1$, and the process continues until the output decision $\beta e(t)O$ is received regarding malicious content from the given link. The encoder is reset to detect malicious content in the next link, and the process is repeated. The encoder possesses updated features $\varepsilon \beta e(t)$, which are important for updating the next layer. Additionally, the encoder is capable of activating $A\beta e(t)$ in the process. The formulation for content detection through the automatic encoder using the GRU is derived as: Here, $t = 0$ initially, and $\beta e(t)O = 0$.

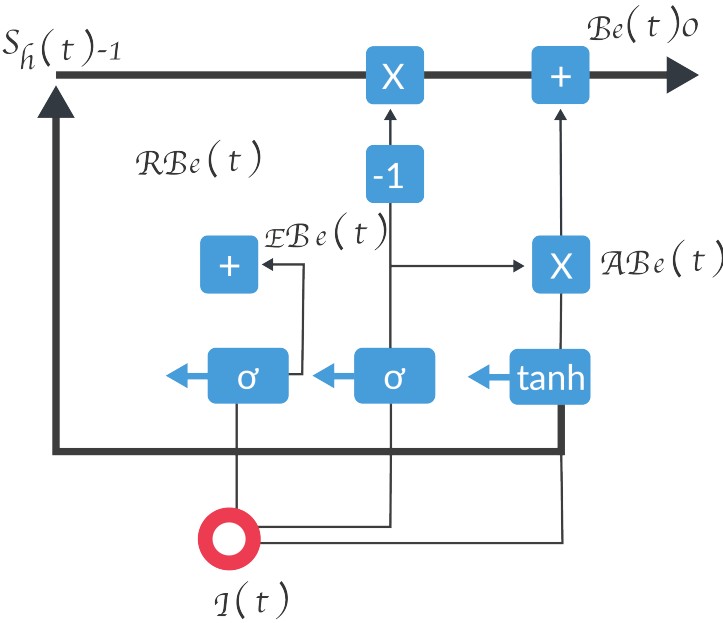

**Figure 7.** Recurrent neural network for malicious/safe content detection from links.

The updated features can be specified as:

$$\varepsilon \beta e(t) = \sigma_{sigmoid}\big\{(\forall \rho \cdot \varepsilon \beta e(t) \cdot l) + \forall \cup \cdot \varepsilon \beta e(t) \cdot S_h(t) - 1) + \forall b \cdot \varepsilon \beta e(t)\big\} \tag{28}$$

$$R\beta e(t) = \sigma_{sigmoid}\big\{(\forall \rho \cdot R\beta e(t) \cdot S_h(t) - 1) + \forall b \cdot \varepsilon \beta e(t)\big\} \tag{29}$$

$$A\beta e(t) = \varphi h\big\{(\beta e(t)O \cdot l) + \forall \cup \cdot A\beta e(t)(R\beta e(t) \oplus S_h(t) - 1) + \forall b \cdot S_h\big\} \tag{30}$$

$$\beta e(t)O = \big\{(1 - \varepsilon \beta e(t)) \oplus (\beta e(t)O - 1) + (\varepsilon \beta e(t) \oplus A\beta e(t))\big\} \tag{31}$$

where $S_h(t)\varphi h$ is a hyperbolic tangent, and $\sigma_{sigmoid}$ is a sigmoid function.

$\forall \rho$, $\forall \cup$, and $\forall b$ are parameters of the autoencoder, which are used to help in detecting the malicious content, given by:

$$\omega = \beta e(l, \forall \rho) \tag{32}$$

The decoder uses $\omega$ as the input to begin the building process. The sequence of the malicious output content $\{i = i^1, i^2, \ldots, i^m\}$ is created by decoder $\beta d$, which is built using another RNN. The hidden state of the decoder is activated with each time step $t$, given by:

$$S_h(t) = f(\beta e)(W_i S_h \cdot S_h(t) - 1, i^t) \tag{33}$$

The overall probability of building the malicious output sequence $i$ producing input link $l$ is defined as:

$$p(i|l; \theta_d) = \prod_{t=1}^{m} p(i_t|i_t - 1, i_t - 2, \ldots, i_1, \omega, \theta_d) \tag{34}$$

where $\theta_d$ is a parameter for the decoder.

To reduce the negative effect of the probability $p$ for all of the malicious content of the link, the components of the autoencoder are mutually trained:

$$l^m(\forall \rho, \forall U, \forall b, \theta_d) = -\sum_{t=1}^{m} log\, p(i_t|l_t; \rho, \forall U, \forall b, \theta_d) \tag{35}$$

## 7. Experimental Setup and Results

To authenticate the performance of the proposed ClickBaitSecurity, realistic testing scenarios were created for malicious link detection and compared with state-of-the-art approaches: LSAC, C-LSTM, and CEM. These experiments were also conducted to determine the advantages of selecting this tool when computing values such as the detection accuracy, system load, malicious and safe links, and the display of a message to the user. To achieve a practical outcome, ClickBaitSecurity was created for detecting malicious links. The Java platform was used to validate the proposed approach. A built-in library that consists of binary search features was used for a faster and efficient search process. A prototype was built to test clickbait.

To conduct the tests, different links, both malicious and safe, were used. A specific link was transferred to the proposed ClickBaitSecurity, which performed the analysis on the obtained data. Two datasets have been used. First consists of headlines from different news sites: https://www.kaggle.com/amananandrai/clickbait-dataset?select=clickbait_data.csv, accessed on 19 November 2021. It includes New York Times, The Guardian, Wikinews, Buzzfeed, The Hindu, ViralStories, Thatscoop, Upworthy, Scoopwhoop, and ViralNova. It involves 32,000 rows that include 50% clickbait and 50% non-clickbait. The second contains 38,517 Twitter posts about 27 major US news publishers: https://zenodo.org/record/5530410#.YcdQBmhBxPY, accessed on 19 November 2021. Furthermore, information about the articles related to the posts is included. Three scenarios were used to measure the performance of ClickBaitSecurity and its counterparts. The minimum system requirements are described in Table 3.

**Table 3.** Basic components for experiments.

| Components | Description |
|---|---|
| Operating system | Windows 10 |
| Processor | Intel(R) Celeron(R) N4020 (1.1 GHz base frequency, up to 2.8 GHz burst) |
| RAM | GB DDR4-2400 MHz RAM (1 × 4 GB) |
| HARD Disk free space | GB DDR4-2400 MHz RAM (1 × 4 GB) |
| Development environment | IntelliJ IDE |

- Scenario 1: ClickBaitSecurity detects the link in a URL blacklist $B_L$ or whitelist $W_L$ and provides the obtained information to the user.
- Scenario 2: ClickBaitSecurity does not detect the link in $B_L$ or $W_L$ but finds its domain name in the domain list and, based on the domain rating of website $D_R$, provides information to the user.
- Scenario 3: ClickBaitSecurity does not find information in any list and provides the user with a message asserting that it cannot analyze this web resource.

Based on the testing process, interesting results for key parameters were obtained:

- Malicious and safe link detection;
- Accuracy.

### 7.1. Malicious and Safe Link Detection

The obtained results are based on three scenarios. The proposed ClickBaitSecurity was able to detect both safe and malicious sites. Additionally, productive informational data on the examined links were received, which also indicates the successful operation of

ClickBaitSecurity. The outcomes of testing the malicious and safe link detection are depicted in Figure 8a,b. Based on the test results, it is observed that the proposed ClickBaitSecurity is capable of detecting more malicious links than the competing methods (LSAC [11], C-LSTM [23], and CEM [25]). This demonstrates that the proposed ClickBaitSecurity avoids returning false negatives.

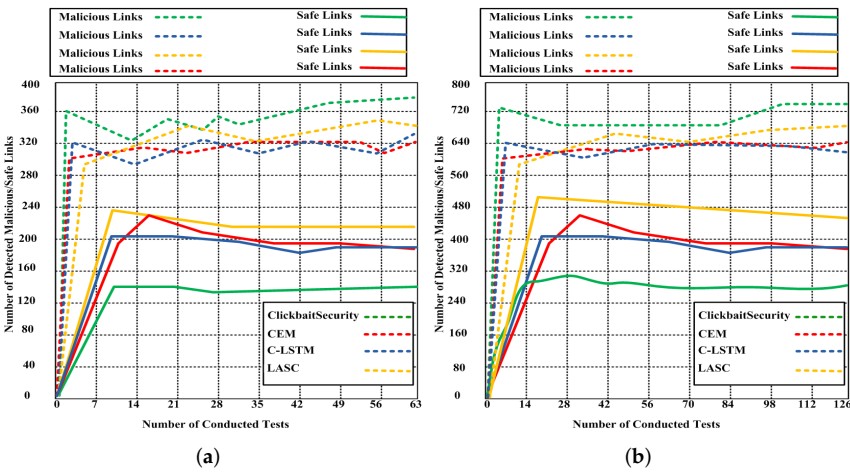

**Figure 8.** (**a**) The number of safe and malicious links detected by the proposed ClickBaitSecurity and competing methods LSAC, C-LSTM, and CEM with a maximum of 63 conducted tests. (**b**) The number of safe and malicious links detected by the proposed ClickBaitSecurity and competing methods LSAC, C-LSTM, and CEM with a maximum of 126 conducted tests.

In 63 conducted tests, ClickBaitSecurity detected 379 malicious links (which were actually malicious links) and 141 safe links, whereas competing approaches, namely, CEM, C-LSTM, and LASC, detected 321, 334, and 346 malicious links, respectively. On the other hand, the competing methods detected more safe links (false negatives) even though some of the links were not safe. CEM is concluded to be a less reliable approach based on its performance. On the other hand, CEM, C-LSTM, and LASC detected 188, 189, and 218 safe links, respectively. In summary, the contending methods incorrectly detected safe links, but they were not safe links. On the other hand, the proposed ClickBaitSecurity detected fewer safe links because the number of the safe links was less that is the reason, the ClickBaitSecurity accurately detected safe links whatever they were available. In short, the proposed ClickBaitSecurity is better than the contending methods by detecting fewer safe links because it avoided false negatives.

When the number of the conducted tests increased, the proposed and compared approaches showed identical performance. Figure 8b shows that the proposed ClickBaitSecurity detected 741 malicious links and 280 safe links with a maximum of 126 conducted tests. On the other hand, C-LSTM, CEM, and LASC detected 621, 640, and 685 malicious links, respectively. The competing approaches showed a similar trend of safe link identification, and some of the links were incorrectly detected as safe (false negatives). CEM, C-LSTM, and LASC detected 388, 389, and 465 safe links, respectively.

Therefore, the number of malicious and safe detected links $D_{ms}$ can be determined as:

$$D_{ms} = D_{lt} \times T_l \times \sum_{i=0}^{L_t} (D_{lc})i + (1 - M_{lt}) \times T_n \tag{36}$$

where $D_{lt}$ is the link detection time, which is the time required to perform the tests and define the nature of the links (either safe or malicious). $T_l$ represents the probability of detecting the trends of links. $D_{lc}$ is the probability of correctly detected links. $M_{lt}$ is the total monitored links. $T_n$ denotes the normal trend of the traffic for monitoring the links.

### 7.2. Accuracy

Accuracy refers to the degree to which the results conform to the truth. Figure 9a–c demonstrate the accuracy of the proposed ClickBaitSecurity and competing approaches (CEM, C-LSTM, and LASC). The results prove that the accuracy of the proposed ClickBait-Security approach is higher than that of competing approaches, as the proposed ClickBait-Security demonstrated 100% accuracy with 450 examined links, as depicted in Figure 9a, whereas CEM, LASC, and C-LSTM achieved 98.04%, 98.25%, and 98.66% accuracy, respectively. When the number of the examined links increased to 900, as depicted in Figure 9b, the accuracy of the proposed ClickBaitSecurity was marginally reduced to 99.95%. On the other hand, the accuracy of the competing approaches also decreased. The accuracy for LASC, CEM, and C-LSTM was determined to be 98.04%, 98.25%, and 98.66%, respectively. When 1800 links were examined, as depicted in Figure 9c, this also had a negative impact on the proposed ClickBaitSecurity and competing approaches. However, the proposed ClickBaitSecurity was only slightly affected, achieving 99.83% accuracy. On the other hand, the competing approaches CEM, LASC, and C-LSTM yielded 96.96%, 97.31%, and 97.39% accuracy, respectively.

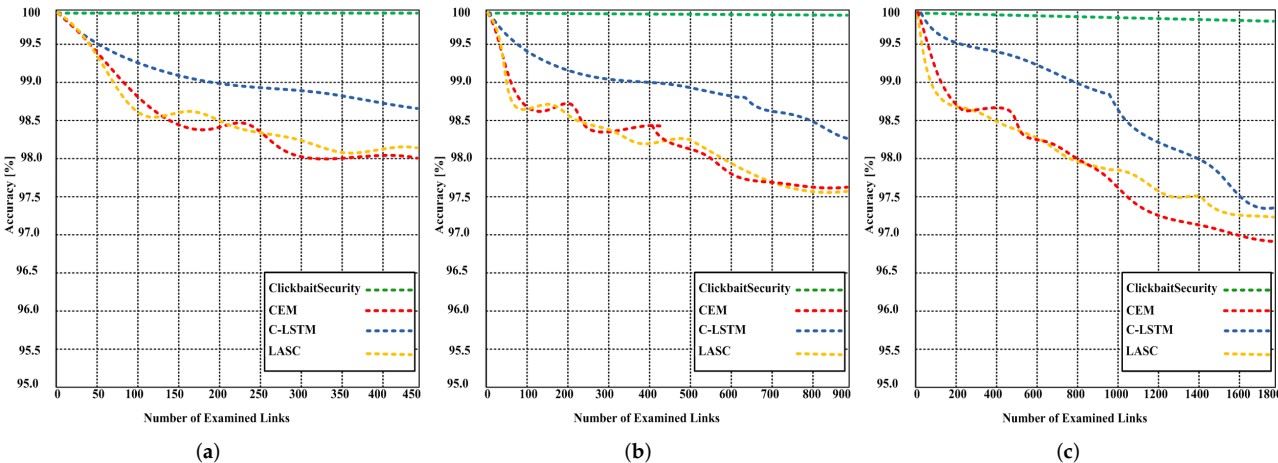

**Figure 9.** (**a**) Accuracy of the proposed ClickBaitSecurity and competing methods with a maximum of 450 examined links. (**b**) Accuracy of the proposed ClickBaitSecurity and competing methods with a maximum of 900 examined links. (**c**) Accuracy of the proposed ClickBaitSecurity and competing methods with a maximum of 1800 examined links.

Thus, the results prove that ClickBaitSecurity was more accurate in detecting clickbait links. The accuracy $A$ of each method is calculated as:

$$A = \frac{P_{mn} + L_m}{P_{mn} + L_{N-m} + L_m + D_{ml}} \tag{37}$$

where $P_{mn}$ is the correct prediction of malicious/non-malicious links, $D_{ml}$ denotes the state in which an existing malicious link in the clickbait is not detected, $L_{N-m}$ represents non-malicious blocked links, and $L_m$ is malicious blocked links. Analyzing the process of clickbait goes through different stages that are the main reason for obtaining higher accuracy. First, the LILS algorithm is used to detect whether the source of the links is white-listed or black-listed. If the sources of the links are found as black-listed, then the black-listed sources are blocked. Later, the focus is only on the white-listed. DRC algorithm is used to make the ranking of the white-listed sources. The white-listed sources with lower-ranking are dropped. Finally, RNN deals with white-listed sources that are of higher rank. Thus, the proposed approach gets higher accuracy with 450 examined links.

## 8. Discussion of Results

The presented ClickBaitSecurity consists of several modules. Modules are responsible for finding links in a whitelist, finding links in a blacklist, and checking the domain rating of a web resource. In general, the work of the extension is simple and inexpensive because it implements the simplest binary search and RNN features.

The results indicate that the created browser extension identifies safe links quite well without blocking them. We also demonstrate that the RNN performs well in identifying malicious links. We can observe that the proposed ClickBaitSecurity shows better results in detecting malicious links than other competing approaches. In Figure 9a–c, the proposed ClickBaitSecurity looks much better in terms of memory usage. Upon completion of testing, suspicious links that led to malicious threats were detected. Based on the results, it is observed that the proposed method does not overload users' personal computers. Our solution has the advantage of analyzing links attached to clickbait and not just blocking them. White- and black-lists of links provide a good level of protection, as they always provide reliable information about web resources. The user can find useful web resources, and legal companies do not lose profits due to the blocking of all advertising. Another advantage of using the proposed ClickBaitSecurity lies in the features of the deep RNN, which helps to successfully differentiate between legitimate and illegitimate links. A further benefit of the proposed ClickBaitSecurity extension is that it assumes that content is malicious until it has complete assurance of its nature; if ClickBaitSecurity is not certain about the content, then alternatively, it provides all of the available information about the content to the user and also provides recommendations for further action.

As the binary sorting method, which is not resource intensive, is used, the user does not need to have a powerful personal computer to use the browser extension. It is also a major advantage for the user that the extension is free to download. If the user has problems during the use of this extension, the user can always leave a review in the Google extension store as feedback for the developer.

However, the proposed ClickBaitSecurity also has a few shortcomings. ClickBaitSecurity can only analyze links within the clickbait, so if it contains malicious executable code, then the extension cannot intervene. If for some reason, the extension cannot obtain sufficient information, then it cannot analyze and provide recommendations to the user. However, the second disadvantage is very unlikely, so users do not need to worry about it. In summary, we conclude that our extension is effective, and anyone with a personal computer can download it and obtain a good level of protection. Table 4 shows the comparative analysis of the proposed ClickBaitSecurity and competing approaches.

**Table 4.** Comparative analysis of the proposed ClickBaitSecurity and competing approaches: CEM, LASC, and C-LSTM.

| Approach | Malicious Link Detection with 63 Conducted Tests | Malicious Link Detection 126 (Conducted Tests) | Safe Link Detection 63 (Conducted Tests) | Safe Link Detection 126 Conducted Tests | Accuracy [%] with 450 Examined Links | Accuracy [%] with 900 (Examined Links) | Accuracy [%] with 1800 (Examined Links) |
|---|---|---|---|---|---|---|---|
| **ClickBaitSecurity** | 9379 Links | 741 Links | 141 Links | 280 Links | 100% | 99.95% | 99.83% |
| CEM | 321 Links | 338 Links | 188 Links | 640 Links | 98.04% | 98.66% | 96.96% |
| C-LSTM | 334 Links | 389 Links | 189 Links | 621 Links | 98.66% | 98.25% | 97.39% |
| LASC | 346 Links | 465 Links | 218 Links | 685 Links | 98.25% | 98.04% | 97.31% |

## 9. Conclusions

Research has demonstrated that clickbait can pose a threat to Internet users. The proposed ClickBaitSecurity assists in evaluating the security of a web resource. Additional information can be obtained using existing sources of information about web resources.

ClickBaitSecurity was developed to solve the security issues of clickbait. A comparison was conducted with existing state-of-the-art approaches (CEM, LASC, and C-LSTM) to determine the effectiveness of the presented solution. The test results demonstrate that the proposed ClickBaitSecurity accurately identifies malicious links and indicates this information to the user, providing a good level of security.

This solution is also optimal from a technical point of view because it is free and not demanding on the user's hardware. Furthermore, the use of the presented browser extension and research into its effectiveness will help to provide a greater level of security. The information technology industry has changed rapidly, and therefore, there is a need to create a mechanism to protect against viruses and threats. Therefore, in the future, we will provide insights into additional metrics.

**Author Contributions:** A.R., conceptualization, writing, idea proposal, methodology, and results; B.A. and M.A., conceptualization, draft preparation, editing, and visualization; S.H., writing and reviewing; A.A. draft preparation, editing, and reviewing. V.J., investigation and reviewing. All authors have read and agreed to this version of the manuscript.

**Funding:** This work was partially supported by the Sensor Networks and Cellular System (SNCS) Research Center under grant 1442-002.

**Institutional Review Board Statement:** Not applicable.

**Informed Consent Statement:** Not applicable.

**Data Availability Statement:** The study did not report any data.

**Acknowledgments:** Taif University Researchers Supporting Project number (TURSP-2020/302), Taif University, Taif, Saudi Arabia. The authors gratefully acknowledge the support of SNCS Research Center at the University of Tabuk, Saudi Arabia. In addition, the authors would like to thank the deanship of scientific research at Shaqra University for supporting this work.

**Conflicts of Interest:** The authors declare no conflicts of interest.

## Appendix A

**Theorem A1.** *For a secure website, the visibility parameter will be low if the web resources are new.*

**Proof.** We have a new website, edo.prgapp.kz (accessed on 19 November 2021). In addition, we have only 2 domains that contain a total of 14 backlinks to this resource, which are calculated by Equation (15). □

**Hypothesis A1.** *If a web resource attached to the clickbait is popular, then the algorithm will be completed in 1 step and will not take much time.*

**Proof.** For example, we consider the Google.com site. It is in the first 100 elements of the whitelist. We can calculate the time to receive a reply message using Equation (1). $T = 0.9$ s. □

Finding this URL in the first 100 elements requires 0.9 s, which is an acceptable time.

**Corollary A1.** *Based on the proof of the first hypothesis, we conclude that the extension requires little time to analyze the most popular web resources, such as Google, since they are detected within the first stage of testing.*

**Hypothesis A2.** *In most cases, the visibility parameter will be very small for new web resources.*

**Proof.** We have the website edo.base.kz (accessed on 19 November 2021). We have only 1 domain that contains a total of 2 backlinks to this resource. The visibility parameter for this resource is equal to $V = 1 \times \sqrt{2} = 1,4$. □

This web resource is safe, but as it is new, it has a low visibility parameter.

**Corollary A2.** *Based on the proof of the second hypothesis, we observe that the visibility parameter is a weak indicator for the analysis of a new web resource. This is because this indicator is based on the number of domains in which links to this web resource are located. Since the new resource has few domains in which links to it are posted, the visibility indicator for new resources will always be small.*

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
