# Peer review of "Clickbait Detection Using Deep Recurrent Neural Network"

_applsci, doi:10.3390/app12010504_

Round 1

Reviewer 1 Report

The authors proposed a novel tool called the ClickBaitSecurity to evaluate the safety of a given link. The system model incorporates the legitimate and illegitimate list search (LILS) algorithm and the domain rating check
(DRC) algorithm where both adopts the binary search features to elevate the efficiency of detecting malicious content. In addition, the ClickBaitSecurity also leverage the features from deep recurrent neural network (RNN) to distinguish between legitimate and illegitimate links. However, I found some problems required clarifications before reconsider it for publication in the Applied Sciences:

  1. The proposed method should try to describe in step-by-step procedure to let the reader easy to understand how it works in establishing and using the model.
  2. Page 7 has some missing references for the algorithm (i.e., "Algorithm ?? ...").
  3. It is unclear how the two algorithms LILS and DRC will work with the results of RNN to complete evaluating the clickbait.
  4. I suggest some of the proofs should move to the appendix as supportive materials.
  5. I did not see a description for the dataset used in this paper. The authors should provide the resources they used for the experiment in order to allow third party to verify the accuracy of their results.
  6. The compared state-of-the-art approaches LSAC, C-LSTM, and CEM have missed to quote the reference numbers. I have to guess their  identifying them via the content in related works.
  7. The accuracy with 450 examined links has reached to 100% for the ClickBaitSecurity. It is hard to believe that a machine learning model can have such perfect result. A 100% accuracy model indicates that there existed potential features that can constitute a rule-based approach to complete the task. Therefore, I suspected the authors might had overfitted their data during the experiment. The authors should re-examined this result and provided some explanations. 

Reviewer 2 Report

This paper presents clickbait detection method by utilizing deep-learning (RNN). The method efficiently evaluated the security of a link based on the legitimate and illegitimate list serach algorithm and the domain rating check algorithm. As discussed in Table 1, there are several related works based on deep learning or a convolutional neural network-based approach for clickbait detection. Both approaches looks familiar so the author should provide detailed comparison on differences. And Table 2 shows the experimental environment. The computer with 200 MB is utilized for testing. It is unclear that RNN requires many learning data and how to learn with only 200 MB disk. Table 3 shows some list of obtained results, but there is no reason to choose these URLs. Lastly, in Table 4, proposed method shows higher malicious link detection rate than others but it shows lower performance on safe link detection. It is hard to say the proposed method is over other methods.

Reviewer 3 Report

The authors proposed an algorithm for the efficient detection of malicious content such as ClickBait links. I have the following observations regarding this paper:

  1. The authors have used many equations (for example eq 1 to eq 26) but it is not clear if these equations are derived by the authors or taken from other works. I suggest citing equations or other authors' work properly.
  2. The dataset used for the result analysis purpose is not discussed properly. It is suggested for a better reading experience, the database used should be explained appropriately.
  3. The novelty of the paper is adequate and the flow of information is excellent.

Round 2

Reviewer 1 Report

I am still not convinced by the Response of the authors in regard to their explanation on the 100% accuracy with 450 examined links. But it is good to see that the authors has revised the manuscript with great efforts to correct other problems pointed out in my previous review report. Therefore, I would like to recommend the manuscript for publication.

Reviewer 2 Report

The paper is updated according to the comment. For this reason, I recommend to accept this paper.